# Paralogue-Specific Roles of SUMO1 and SUMO2/3 in Protein Quality Control and Associated Diseases

**DOI:** 10.3390/cells13010008

**Published:** 2023-12-20

**Authors:** Wei Wang, Michael J. Matunis

**Affiliations:** Department of Biochemistry and Molecular Biology, Bloomberg School of Public Health, Johns Hopkins University, Baltimore, MD 21205, USA; wwang107@jhmi.edu

**Keywords:** SUMO, protein quality control, neurodegenerative disease, cardiovascular disease, cystic fibrosis, proteostasis

## Abstract

Small ubiquitin-related modifiers (SUMOs) function as post-translational protein modifications and regulate nearly every aspect of cellular function. While a single ubiquitin protein is expressed across eukaryotic organisms, multiple SUMO paralogues with distinct biomolecular properties have been identified in plants and vertebrates. Five SUMO paralogues have been characterized in humans, with SUMO1, SUMO2 and SUMO3 being the best studied. SUMO2 and SUMO3 share 97% protein sequence homology (and are thus referred to as SUMO2/3) but only 47% homology with SUMO1. To date, thousands of putative sumoylation substrates have been identified thanks to advanced proteomic techniques, but the identification of SUMO1- and SUMO2/3-specific modifications and their unique functions in physiology and pathology are not well understood. The SUMO2/3 paralogues play an important role in proteostasis, converging with ubiquitylation to mediate protein degradation. This function is achieved primarily through SUMO-targeted ubiquitin ligases (STUbLs), which preferentially bind and ubiquitylate poly-SUMO2/3 modified proteins. Effects of the SUMO1 paralogue on protein solubility and aggregation independent of STUbLs and proteasomal degradation have also been reported. Consistent with these functions, sumoylation is implicated in multiple human diseases associated with disturbed proteostasis, and a broad range of pathogenic proteins have been identified as SUMO1 and SUMO2/3 substrates. A better understanding of paralogue-specific functions of SUMO1 and SUMO2/3 in cellular protein quality control may therefore provide novel insights into disease pathogenesis and therapeutic innovation. This review summarizes current understandings of the roles of sumoylation in protein quality control and associated diseases, with a focus on the specific effects of SUMO1 and SUMO2/3 paralogues.

## 1. Introduction

Proteostasis is critical for cell heath and normal cellular functions. Perturbation of proteostasis results in accumulation of cytotoxic aberrant proteins, which is associated with a spectrum of human diseases including cardiovascular disease [1] and neurodegeneration [2]. Maintenance of proteostasis relies on protein quality control (PQC), cellular surveillance systems that monitor proper folding of native proteins and removal of misfolded and damaged proteins. PQC networks are intimately associated with post-translational modifications (PTMs), with ubiquitylation playing a central role. Although identified as members of the ubiquitin-like protein family, SUMOs were initially thought to carry out anti-ubiquitin functions, as the two modifiers can compete for the same lysine residues within substrate proteins [3]. Moreover, early characterized functions of sumoylation were mostly associated with genome integrity and modulation of protein–protein interactions, without a direct connection to protein degradation. This perspective was overthrown by the discovery of SUMO-targeted ubiquitin E3 ligases (STUbLs), which revealed that poly-SUMO2/3 modification can serve as a signal for ubiquitylation and subsequent proteasomal degradation of target proteins [4]. The prototypical STUbL, RNF4, was soon found to be involved in many of the previously identified nuclear functions of SUMO2/3, suggesting that mediating protein turnover can be a major function of these paralogues [5]. Recent studies have also revealed functions for SUMO1 in modulating protein solubility and stability without clear involvement of STUbLs [6,7]. Furthermore, sumoylation also directly regulates PQC factors, including molecular chaperones and components of the proteasome [8,9,10]. Emerging studies have linked both SUMO1 and SUMO2/3 to human diseases whose pathogeneses are associated with disturbed proteostasis and cytotoxic protein aggregation, such as cardiomyopathy and nearly all forms of neurodegeneration [11,12]. Targeting sumoylation therefore represents a potential therapeutic strategy to treat or limit the progression of these diseases. A better understanding of SUMO paralogue-specific effects may provide novel angles for the development of targeted therapies.

In the current review, we discuss unique properties of SUMO1 and SUMO2/3 and existing evidence of paralogue-specific functions in PQC and PQC-associated diseases. Based on current knowledge, we propose a general model where SUMO2/3 functions in concert with STUbLs and the ubiquitin–proteasome system to promote substrate degradation, while SUMO1 modification promotes solubility of aberrant proteins and inhibits aggregation.

## 2. The SUMO Family

The small ubiquitin-related modifiers (SUMOs) comprise a family of post-translational protein modifiers that are ~11 kDa in size and structurally resemble ubiquitin. However, they share only ~20% amino acid sequence identity with ubiquitin and carry out distinct cellular functions (Figure 1A,B). SUMO proteins are expressed across the eukaryotic kingdom. In yeast, *C. elegans* and *Drosophila*, a single SUMO protein is expressed, whereas vertebrates express multiple SUMO paralogues [13]. In humans, genes coding for five SUMO paralogues, referred to as SUMO1, SUMO2, SUMO3, SUMO4 and SUMO5, have been identified. SUMO1, SUMO2 and SUMO3 are ubiquitously expressed and are the best studied paralogues, whereas SUMO4 and SUMO5 show low, tissue-specific expression and their functions remain enigmatic [14,15,16,17]. Mature forms of SUMO2 and SUMO3 share ~97% sequence identity and are commonly referred to as SUMO2/3, but they share only ~47% sequence identity to SUMO1 [18] (Figure 1A,B). SUMO4 is most similar to SUMO2/3 but contains a unique proline residue (P90) at the C-terminus that prevents efficient maturation [19] (Figure 1A). Nonetheless, genetic studies have linked SUMO4 to both type 1 and 2 diabetes, suggesting a possible role for non-conjugated SUMO4 [20,21]. The most recent addition to the family, SUMO5, may be a pseudogene of SUMO1 but has nevertheless been shown to regulate dynamics of PML nuclear bodies when expressed exogenously in cells [16]. Whether SUMO5 is endogenously expressed is unclear. Defining the biological functions and mechanisms of action of SUMO4 and SUMO5 require further investigations.

Gene knockout experiments revealed essential roles for sumoylation in *S. cerevisiae* [22], *C. elegans* [23], *A. thaliana* [24] and mice [25]. In mice, SUMO2 is essential while SUMO3 is dispensable for embryonic development. These differences in essential function are very likely related to differences in expression levels, with SUMO2 being expressed at much higher levels relative to SUMO3 [25]. Requirements for SUMO1 in embryonic development are potentially context dependent, with different knockout studies reaching different conclusions [26,27]. Nonetheless, loss of SUMO1 leads to severe post-developmental phenotypes in mice, including increased incidence of congenital heart disease, cleft lip and palate and altered response to high-fat diet [26,28,29].

## 3. The Sumoylation Pathway

SUMOs are expressed as precursor proteins and require proteolytic maturation by SUMO-specific proteases to expose a C-terminal di-glycine motif prior to conjugation [13]. Although conserved from yeast to human and across paralogues, the biological significance of SUMO precursors remains uncertain. One possible explanation is that they serve a reservoir of inactive, free SUMO proteins that can be rapidly processed and conjugated to substrate proteins during stress responses. Mature forms of SUMOs are conjugated to substrate proteins through a three-step enzymatic cascade that is analogous to ubiquitylation (Figure 2). Multiple factors within the sumoylation pathway, including the E2 conjugating enzyme Ubc9 and several E3 ligases, have been implicated in cellular PQC and associated diseases, as discussed in the following sections.

Mature forms of SUMOs are first activated in an ATP-dependent manner by the E1 activating enzyme heterodimer, Sae1/Sae2, followed by transfer to the catalytic cysteine residue of the single E2 conjugating enzyme, Ubc9. In the final step of sumoylation, SUMO is transferred from Ubc9 to a lysine residue within substrate proteins. While for ubiquitylation, an E3 ligase is required for substrate selection and conjugation, sumoylation may occur independent of E3 ligases through direct recognition of a SUMO consensus sequence conjugation site (Y-K-X-E) within substrates by Ubc9 [30,31,32]. Ubiquitin E3 ligases bind to both ubiquitin-charged E2 enzymes and substrate proteins, thereby conferring substrate selectivity [33]. Mechanisms that mediate substrate selection for sumoylation are not well understood. Whether all SUMO E3s bind directly to substrate proteins also remains uncertain. In a widely accepted model, the SUMO-Ubc9 thioester is pre-oriented for target binding, while E3 ligases stabilize the optimal conformation for catalysis and thereby enhance conjugation efficiency [32]. Compared to >600 identified ubiquitin E3 ligases within humans, only a handful of SUMO E3 ligases have been characterized to date (Table 1) [34]. The stark contrast between the limited number of SUMO E3 ligases and thousands of identified sumoylation substrates suggests that SUMO E3s are more promiscuous in substrate selection, with each E3 mediating SUMO conjugation to a range of cellular proteins. Previously, the tripartite motif (TRIM) protein family member PML (TRIM19) was shown to recognize misfolded nuclear proteins through a coiled-coil region within its TRIM/RBCC motif, followed by sumoylation of these proteins through its SUMO ligase activity [35]. This observation is consistent with PML having a promiscuous mechanism of substrate selection and a broad function in nuclear PQC.

In yeast, four major SUMO E3 ligases have been identified: Siz1, Siz2, Mms21 and Zip3 [36]. In humans, SUMO E3 ligase activity has been found in members of the protein inhibitor of activated STAT (PIAS) family and the TRIM superfamily [37,38]. Both classes of proteins impart E3 ligase activity through a really interesting new gene (RING) domain typically found within ubiquitin and SUMO E3 ligases. Several noncanonical ligases that function independent of RING domains have also been identified, including nucleoporin Nup358 (RanBP2), the polycomb group protein PC2, the topoisomerase I-binding RING finger protein (TOPORS) and the SUMO2/3-specific E3 ligase RNF451 [39,40,41,42]. Understanding how these SUMO E3 ligases confer substrate specificity, and how selective conjugation of specific SUMO paralogue to certain substrates is achieved, require further investigations. The determinants of mono- versus poly-SUMO modifications are also unclear and warrant future explorations.

**Table 1 cells-13-00008-t001:** Human SUMO E3 ligases.

Type	Protein	Reference
PIAS protein family	PIAS1	Kahyo et al., 2001 [43]
	PIAS3	Nakagawa and Yokosawa, 2002 [44]
	PIASxα	Nishida and Yasuda [45]
	PIASxβ	Schmidt and Müller, 2002 [46]
	PIAS4 (PIASγ)	Galanty et al., 2009 [47]
TRIM protein superfamily	TRIM19 (PML)	Guo et al., 2014 [35]
	TRIM27	Chu and Yang, 2010 [38]
	TRIM28 (KAP1)	Li et al., 2020 [48]
	TRIM 32	Chu and Yang, 2010 [38]
Noncananical SUMO E3s	RanBP2	Pichler et al., 2004 [39]
	PC2	Kagey et al., 2003 [40]
	TOPORS	Weger et al., 2005 [41]
	Rhes	Subramaniam et al., 2010 [49]
	RNF451	Koidl et al., 2016 [42]

## 4. Dynamic SUMO Conjugation and Deconjugation Equilibrium

Sumoylation can be reversed by specific isopeptidases that remove and recycle conjugated SUMOs from target proteins [50]. Three distinct types of cysteine isopeptidases have been identified for SUMO deconjugation: (1) the Ulp/SENP (ubiquitin-like protease/sentrin-specific protease) family, (2) the Desi (de-sumoylation isopeptidase) family and (3) USPL1 (ubiquitin-specific peptidase-like protein 1). In humans, six SENPs have been identified (SENP1-3 and SENP5-7) with different SUMO paralogue preferences. SENP1 and SENP2 deconjugate all SUMO isoforms, but SENP1 demonstrates a slight preference for SUMO1 and SENP2 shows a slight preference for SUMO2, while SENP3 and SENP5-7 all preferentially deconjugate SUMO2/3 [34]. Apart from deconjugating SUMOs from substrates, SENPs also mediate maturation of SUMO precursors by removing C-terminal amino acids to expose the di-glycine motif necessary for conjugation (Figure 2). Compared to the SENP family, functions of Desi and USPL1 are less defined. Desi appears to only exert isopeptidase activity on selected substrates, with transcriptional repressor BZEL being the only identified substrate to date [51]. USPL1 was identified in an activity-based search for novel SUMO isopeptidases [52]. Although shown to be associated with Cajal bodies within the nucleus, endogenous substrates of USPL1 remain unidentified.

Due to competing actions of conjugating enzymes and isopeptidases, sumoylation is highly dynamic in cultured mammalian cells. At steady state, the majority of SUMO1 is conjugated to proteins, with Ran GTPase-activating protein1 (RanGAP1) being a major reservoir for SUMO1 conjugation [53]. In contrast, a considerable pool of unconjugated SUMO2/3 is maintained within cultured cells, which can be rapidly conjugated to proteins in response to cellular stresses such as heat shock and proteasomal inhibition [54,55,56]. It has therefore been proposed that SUMO2/3 conjugation serves as a rapid protective cellular response under stress. Extensive crosstalk between ubiquitylation and SUMO2/3 modification upon proteasome inhibition has been revealed through sequential immunopurification and mass spectrometry analysis, supporting roles for SUMO2/3 in cellular PQC against proteotoxic stress [10]. Furthermore, increased global SUMO2/3 conjugation has also been shown to work synergistically with the HSF1-regulated chaperone network to promote proteostasis under heat shock [56].

## 5. Poly-SUMO Chains and SUMO-Interacting Motifs

Similar to ubiquitin, SUMO2/3 can form polymeric chains through a conserved lysine (K11) residue embedded within a consensus modification site in the N-terminus [57,58]. In contrast, SUMO1 lacks consensus modification sites and has been proposed to modify proteins either as a monomer or serve as a terminator of poly-SUMO2/3 chains [59,60]. However, these views are challenged by findings from recent proteomic analyses. While K11 is the most frequently modified residue for poly-SUMO2/3 chain formation in cultured human cells, in mouse organs, K21 and K33 are the most frequently modified residues [61]. Poly-SUMO2/3 chains were also detected on K7 of SUMO1, which is located within an inverted consensus sumoylation site [61]. Chain formation through different lysine residues may result in different chain topologies that serve as unique cellular signals, as exemplified by K48- and K63-linked polyubiquitin chains, which function as signals for proteolysis and DNA damage repair, respectively [62,63]. Although studies in *S. cerevisiae* revealed no evidence for linkage-specific SUMO chain functions [64], possible functions in mammalian cells have not been fully investigated.

In addition to being covalently conjugated to target proteins, SUMOs can also interact with proteins through noncovalent association with SUMO-interacting motifs (SIMs), which comprise a short hydrophobic stretch of amino acids (V/I-X-V/I-V/I, where X is any amino acid) flanked on either side by acidic residues [65]. This motif can embed itself into a groove formed between the α-helix and β2-strand on the surface of SUMO, forming a short parallel or antiparallel β-strand configuration (Figure 3) [65,66]. Modulating protein–protein interactions through SUMO–SIM binding represents a major mechanism by which sumoylation regulates diverse cellular processes. A classic example is the assembly of PML nuclear bodies (PML-NBs), where sumoylation of PML protein facilitates the recruitment of SIM-containing PML-NB-associated proteins [67]. Some SIMs are selective for specific SUMO paralogues due to amino acid variation within the binding interface, leading to paralogue-selective interactions between SUMO-modified and SIM-containing proteins [68]. Furthermore, SIMs may also direct paralogue-specific substrate modification. As examples, USP25 and the Bloom syndrome DNA helicase (BLM) are preferentially modified by SUMO2/3 due to higher affinity of their SIMs for SUMO2/3-charged Ubc9 [69,70].

The ability of SUMOs to affect proteins through both covalent conjugation and noncovalent interaction enables them to regulate a vast landscape of the proteome [61,71]. The possibility to impart paralogue-specific regulation of substrate proteins further confers sumoylation the power to achieve sophisticated, multilevel regulation in cellular pathways. Under the context of cellular PQC, poly-SUMO2/3 modification can serve as a signal for STUbL-mediated ubiquitylation and subsequent proteasomal degradation [72], while SUMO1 may exert chaperone-like functions, either through covalent conjugation or non-covalent SIM-binding, to promote substrate solubility [73,74]. As SIMs comprise a stretch of hydrophobic residues, misfolded proteins may directly attract free SUMOs through interactions with exposed hydrophobic regions that serve as SIMs (Figure 5). The SUMO protein could thereby mask the exposed hydrophobic patches to maintain solubility of misfolded substrates and prevent aggregation before engagement by molecular chaperones and other PQC factors. The pioneer SUMO protein may also serve as a mediator to recruit Ubc9 and E3 ligases that subsequently modify misfolded substrates and direct their degradation through the STUbL pathway.

## 6. The Sumoylated Proteome

The dynamic conjugation and deconjugation equilibrium in cells results in low steady-state levels of individual sumoylated proteins, making it challenging to detect endogenous substrates experimentally [75]. Nonetheless, recent advances in proteomic analysis have led to identification of thousands of putative sumoylation substrates, with the most in-depth analysis targeting the SUMO2-modified proteome [61]. Gene ontology analysis of identified substrates has revealed the most frequently detected sumoylation targets, including transcription factors, chromatin regulators, factors involved in DNA damage response and cell cycle regulators [76]. In addition, functions of sumoylation outside the nucleus are also evidenced by modification of ion channels [77], cytoskeletal proteins [78], factors in the autophagy pathway [79] and mitochondrion-associated proteins [80]. Proteomic analysis revealed a considerable overlap of sumoylated and ubiquitylated substrates and modification sites, demonstrating extensive interplay between the two modifications (Figure 4A,B) [10,81,82]. Global ubiquitylation and SUMO2/3 conjugation levels increase upon proteasome inhibition and heat stress, further pointing to cooperative roles of the two pathways in response to proteotoxic stress [83]. Interestingly, increased SUMO2/3 conjugation induced by heat shock and proteasome inhibition depends mainly on new protein synthesis, suggesting functions for SUMO2/3 in regulating newly synthesized unfolded and misfolded proteins under proteotoxic stress [83,84]. Table 2 lists major proteomic studies conducted in recent years and the number of sumoylation substrates/sites identified in each study. Notably, existing large-scale proteomic analyses have focused primarily on SUMO2/3 modification, while a side-by-side comparison of SUMO1 and SUMO2/3 conjugated proteomes using more recently developed, robust high-throughput technologies has yet to emerge. Furthermore, existing analyses rely heavily on overexpression of exogenously introduced epitome-tagged SUMO proteins, which may obscure paralogue-specificity.

Evidence from studies of specific substrate proteins has revealed that modification of SUMO1 and SUMO2/3 can lead to distinct functional outcomes. For instance, SUMO1 and SUMO2 differentially regulate the stability and function of circadian protein Period2 (PER2). While SUMO2 modification promotes proteasomal degradation of PER2, SUMO1 conjugation inhibits its degradation and enhances its function as a transcriptional suppressor [85]. Similarly, poly-SUMO2/3 modification of mutant CFTR protein associated with cystic fibrosis promotes its degradation, whereas SUMO1 conjugation enhances CFTR stability [86]. These compelling observations underscore the necessity of further explorations of SUMO paralogue-specific effects on individual substrate proteins. The dynamics and possible temporal control of functionally distinct SUMO1 and SUMO2/3 modifications in cellular processes and during stress responses also warrant further investigations.

**Table 2 cells-13-00008-t002:** Recent proteomic analyses of the sumoylated proteome.

Reference	Number of Identified Substrates/ Sumoylation Sites	SUMO Paralogue Investigated	Treatment/ Conditions
González-Prieto et al., 2021 [71]	379 non-covalent SUMO interactors in HeLa cells	Monomeric SUMO1 and SUMO2; trimeric SUMO2 chains	In vitro binding of whole-cell lysate to His-SUMO immobilized on Ni-NTA beads
Hendriks et al., 2018 [61]	14,869 endogenous sites mapped to 3870 proteins in HEK293 cells; 1963 conjugation sites in 8 mouse tissues	SUMO2	Heat stress and proteasome inhibition
Hendriks et al., 2017 [76]	40,765 sumoylation sites mapped to 6747 human proteins	SUMO2	Native condition and proteasome inhibition
Lumpkin et al., 2017 [87]	1209 endogenous sumoylation sites	All paralogues	Native condition and proteasome inhibition
Lamoliatte et al., 2017 [10]	10,388 sumoylation sites in HEK293 cells	SUMO3	Exogenous expression of His-SUMO3; Proteasome inhibition
Hendriks and Vertegaal, 2016 [88]	A database that summarizes 22 human SUMO proteomic studies as of 2016. In total, 3617 sumoylated proteins and 7327 sumoylation sites were identified	-	Various conditions

## 7. Proteostasis and the Importance of Protein Quality Control

Generally, cellular PQC employs three distinct yet interconnected strategies whereby misfolded proteins can be refolded, degraded or sequestered into specific cellular compartments that separate them from the intracellular environment [89]. Molecular chaperones play a central role in determining the fate of misfolded proteins inside the cell, while clearance of terminally misfolded proteins is accomplished by two major proteolytic machineries: the ubiquitin–proteasome system (UPS) and autophagy-lysosomal degradation [89,90]. Furthermore, distinct and interconnected PQC pathways are spatially organized in different cellular compartments, ensuring efficient clearance of misfolded proteins at different cellular locations [91,92]. The following sections summarize major cellular PQC pathways with an emphasis on UPS-associated mechanisms, and what is known about sumoylation in each process.

## 8. Molecular Chaperones

Molecular chaperones make up ~10% of the proteome and are essential components of PQC [93]. The majority of molecular chaperones are called heat-shock proteins (HSPs), for their expression can be induced by cellular stresses such as heat shock and oxidative stress. HSPs are further divided into several subgroups based on their molecular sizes, such as Hsp90, Hsp70, Hsp60/chaperonins, Hsp40 (DnaJ) and small HSPs (sHsps). Chaperones recognize unfolded nascent proteins and misfolded proteins through exposed hydrophobic regions and can assist their refolding using energy provided by ATP hydrolysis (with sHsps, which work in an ATP-independent manner, being an exception) [94,95]. If refolding fails, chaperones can facilitate the degradation of terminally misfolded proteins in cooperation with cellular proteolytic systems. For instance, ubiquitin E3 ligases that mediate proteasomal degradation usually depend on chaperones for substrate recognition [90]. Moreover, some misfolded proteins carrying the KFERQ sequence can be recognized by specific chaperones and directly delivered to the lysosome through chaperone-mediated autophagy (CMA) [90]. The decision between refolding and delivery to downstream degradation machineries is further mediated by binding of different co-chaperones [96]. Some chaperones, such as hexametric AAA-ATPases, can also function as “disaggregases” that break down protein aggregates by extracting proteins in an ATP-dependent manner [97,98].

Enhanced SUMO2/3 conjugation upon heat shock was previously shown to act in concert with the heat shock transcription factor 1 (HSF1)-regulated chaperone network to reduce protein aggregation and maintain proteostasis under stress [56]. Chaperone depletion led to prolonged SUMO2/3 conjugation and impaired clearance of ubiquitin-SUMO2/3 co-modified substrates during stress recovery, suggesting an interplay between sumoylation and the cellular chaperone network in mediating protein degradation upon proteotoxic stress [56]. HSF1 itself is modified by SUMO1, and sumoylation may impact its activity during stress response [99]. Multiple chaperones are also directly modified and regulated by sumoylation, including Hsp90 [8] and the AAA-ATPase VCP/p97 [9]. Sumoylation of Hsp90 facilitates its binding to both the activating co-chaperone Aha1 and ATP-competitive Hsp90 inhibitors [8,100]. In the case of p97, sumoylation of the protein itself promotes hexamer assembly and regulates its subcellular distribution to the nucleus and stress granules [9]. Furthermore, the p97 cofactor Ufd1 contains a SIM, which promotes recruitment of the Ufd1-Npl4-p97 disaggregase complex to sumoylated proteins, such as RAD52 under the context of DNA damage [101].

## 9. The Ubiquitin–Proteasome System (UPS)

The UPS is a major component of PQC that mediates the degradation of >80% of cellular proteins [90]. Protein degradation through the UPS requires two successive steps: covalent attachment of multiple ubiquitin moieties to substrate proteins as polymeric chains (a classical proteasomal degradation signal comprises ≥4 ubiquitin molarities linked by internal Lys48 residues), followed by proteolysis of ubiquitylated substrates by the 26S proteosome. Delivery of ubiquitylated substrates to the 26S proteasome is facilitated by molecular chaperones [96]. The 26S proteasome is composed of a barrel-shaped 20S core complex capped on one or both sides by the 19S regulatory complex. The latter is responsible for recognizing polyubiquitin signals, removing and recycling free ubiquitin from substrate proteins and translocation of unfolded substrates into the catalytic core [102]. The 20S proteolytic core degrades unfolded substrates into short peptides of 2–24 amino acids, which can be further degraded into free amino acids by aminopeptidases [103]. Several proteasome subunits are extensively sumoylated with a majority of modified lysines located on the surface of the 20S subunit, suggesting that sumoylation may influence the sub-cellular localization of proteasomes through SUMO–SIM interactions [10]. Supporting this view, the SIM motif within PML IV protein is required for efficient recruitment of proteasomes to PML nuclear bodies [10].

Crosstalk between sumoylation and the UPS can happen on multiple levels (Figure 4B). First, SUMO and ubiquitin can modify the same lysine residue within substrate protein, leading to different fates for the modified protein. The majority of currently identified sumoylation sites are also acceptors for ubiquitylation, and nearly all identified sumoylation substrates can also be ubiquitylated (Figure 4A). However, overlap in conjugation sites within substrate proteins does not necessarily lead to competition between the two modifications. Instead, sequential modification by SUMO and ubiquitin can also convey cooperative regulation of substrate proteins. For example, under genotoxic stress, sumoylation promotes nuclear translocation and subsequent ubiquitylation of NF-*κ*B essential regulator (NEMO) on the same lysine residue [104]. Ubiquitylated NEMO is ultimately transported back into the cytosol for NF-*κ*B activation. Apart from exclusive sumoylation and ubiquitylation on the same lysine, poly-SUMO2/3 modification can also serve as a signal to recruit STUbLs, which subsequently ubiquitylates poly-SUMO2/3 chains and/or other lysine sites within substrate proteins, leading to downstream proteasomal degradation. Last but not least, sumoylation may also directly regulate activity of the proteasome and its recruitment to sites of active protein degradation.

## 10. STUbLs and a SUMO-Mediated Nuclear PQC Pathway

The prototypical mammalian STUbL, RNF4, was first discovered as a co-regulator of transcription mediated by steroid receptors [105]. Its yeast ortholog, the slx5/slx8 heterodimer, was identified in a screen for genes that confer synthetic lethality with the Sgs1 DNA helicase (Bloom syndrome protein (BLM) in humans) [106]. RNF4 has since been shown to regulate diverse cellular functions, including DNA replication [107], DNA damage response [108], nuclear protein quality control [35] and regulation of subcellular structures, such as PML-nuclear bodies [4], by coordinating the sumoylation and ubiquitylation pathways. Functions of RNF4 can be counteracted by ubiquitin-specific protease 7 and 11 (USP7 and USP11), which deubiquitylate hybrid SUMO–ubiquitin chains [109,110]. RNF4 contains four SIMs arranged in tandem, leading to preferential targeting of proteins modified by poly-SUMO2/3 chains [4]. A second human STUbL, Arkadia/RNF111, was identified in a computational screen for proteins containing clusters of SIMs as in RNF4 [60]. In addition to two SIMs, RNF111 also contains a SUMO1-binding motif and shows a strong binding preference for substrates bearing SUMO1-capped SUMO2/3 chains [60]. Compared to RNF4, functions of RNF111 are less defined, although it has been shown to regulate DNA damage response [111] and the TGF-β signaling pathway [112]. RNF4 and RNF111 are both predominantly nuclear proteins and regulate nuclear pathways [111,113]. Whether the two proteins also function outside the nucleus, and whether other unidentified STUbLs play dominate roles in other cellular compartments, are unclear and require future investigations.

RNF4 was previously shown to work in concert with PML to promote clearance of misfolded nuclear proteins. In this pathway, misfolded proteins are selectively recognized by PML and conjugated with poly-SUMO2/3 chains through its SUMO E3 ligase activity [35]. Poly-SUMO2/3 modified proteins are subsequently recognized and ubiquitylated by RNF4 and targeted for proteasomal degradation. This pathway was further shown to promote degradation of a broad range of pathogenic proteins associated with multiple neurodegenerative diseases, including Ataxin-1, huntingtin, and TAR DNA-binding protein 43 (TDP-43) [35]. The SUMO2/3-RNF4-UPS degradation pathway is also involved in the homeostasis of PML-NBs, as PML itself and multiple PML-NB-associated proteins are sumoylated [114]. Functions of RNF4 in genome stability and maintenance have also been extensively explored in the past decades and reviewed previously [5]. In a proteomic search for endogenous poly-sumoylated substrates, >300 proteins involved in diverse cellular functions were identified as potential RNF4 substrates [115], suggesting that the SUMO2/3-RNF4 pathway may be a general pathway for degradation of cellular proteins. Interestingly, Ubc9 and several SUMO E3 ligases are RNF4 substrates [116], suggesting that RNF4 may also negatively regulate sumoylation and SUMO-dependent protein degradation.

## 11. SUMO-Regulated PQC Outside the Nucleus

In addition to the well-defined roles of poly-SUMO2/3 and RNF4 in nuclear PQC, recent findings in our lab revealed that SUMO1 also imparts unique functions in cytosolic PQC. Using a well-characterized PQC model substrate, GFP-CL1, we found that SUMO1 uniquely promotes solubility and degradation of this unstable cytosolic protein, while SUMO2 does not [6,117]. Knockdown of RNF4 by siRNA revealed that SUMO1 promotes substrate degradation independently of RNF4. Mechanistically, we hypothesized that SUMO1 promotes solubility and prevents aggregation of misfolded proteins by intramolecular association with exposed hydrophobic residues serving as SIMs. Furthermore, SUMO1 conjugation may also direct substrate targeting to downstream degradation machineries through SUMO–SIM interactions with PQC pathway components. Supporting this view, we found that GFP-CL1 is conjugated by SUMO1 and SUMO1 is required for efficient downstream ubiquitylation of the protein [6]. For the GFP-CL1 substrate, the observed effects of SUMO1 were restricted to the cytosol but not in the nucleus. Whether this is due to function of particular E3 ligases that localize specifically within the cytosol, or interactions with other cytosolic PQC factors, is unclear and requires further investigation.

In addition to regulating stability of nuclear and cytosolic proteins, limited understandings have also been gained on the effects of sumoylation on proteins associated with membrane-bound organelles. Sumoylation levels of mitochondrial proteins are enhanced upon import failure and impaired proteostasis, suggesting a role for sumoylation in quality control of mitochondrial proteins [118]. Furthermore, an unbiased CRISPRi screen identified SUMO E1 subunit Sae1 as a key regulator of membrane protein targeting to the endoplasmic reticulum (ER) and mitochondria [119]. Knockdown of Sae11 resulted in mislocalization and reduced abundance of multiple mitochondrial tail-anchored proteins, which may be explained by destabilization of mistargeted tail-anchored proteins [119].

## 12. A Generalized Model for SUMO1- and SUMO2/3-Mediated PQC Pathways

Based on current evidence, we propose a model for the general functional mechanisms of SUMO1 versus SUMO2/3 in cellular PQC (Figure 5). In this model, poly-SUMO2/3 modification of aberrant proteins recruits STUbLs, which subsequently lead to degradation through the UPS. The SUMO2/3–STUbL pathway appears to be a general cellular PQC mechanism based on findings that global SUMO2/3 conjugation levels increase in response to proteotoxic stress and that ubiquitin is associated with purified SUMO2 conjugates but not SUMO1 conjugates [56,81]. On the other hand, SUMO1 modification functions to maintain unfolded or misfolded proteins in the soluble form, thereby preventing aggregation and facilitating recognition by cellular PQC machineries through SIM interactions. Supporting this model, SUMO1 has been previously shown to improve solubility of α-synuclein, the major pathogenic protein associated with Parkinson’s disease, and prevent its aggregation both in vitro and in vivo [7,73]. Moreover, an unconjugatable SUMO1-derived peptide inhibitor comprising amino acid 15-55 (SUMO1(15-55)) attenuates α-synuclein aggregation by binding to two SIMs within its aggregation-regulating regions [120]. The effect of SUMO1 modification on the stability of α-synuclein was not evaluated in these studies.

It should be noted that our proposed working model is not inclusive of all substrates and scenarios, and exceptions to the model will exist. As an example, STUbL protein RNF111 preferentially recognizes substrates modified by SUMO1-capped poly-SUMO2/3 chains, suggesting that SUMO1 also participates in modulating protein degradation [60]. Furthermore, while lacking direct evidence, it is possible that multiple mono-SUMO1 modifications may be sufficient to recruit STUbL proteins and induce degradation.

The downstream consequence of SUMO1 modification and enhanced protein solubility can vary in a context-dependent manner as well. On one hand, SUMO1 may compete with poly-SUMO2/3 modification, thereby blocking the SUMO2/3–STUbL pathway and ubiquitin-mediated degradation, leading to enhanced stability of substrate proteins. A good example for this scenario is paralogue-specific regulation of CFTR homeostasis, as discussed in the next section. On the other hand, SUMO1-mediated enhancement in solubility may improve accessibility to other degradation pathways and thereby promote protein turnover, as demonstrated for GFP-CL1 [6]. How the cell regulates different impacts of SUMO1 modification on substrate protein quality control is unclear. What defines substrate specificity for SUMO1- and SUMO2/3-mediated PQC pathways and what are the determinants of SUMO1 versus SUMO2/3 modification of specific substrate proteins are also unclear and require further investigations.

## 13. PQC-Associated Diseases—A SUMO Perspective

Targeting proteolysis is an intensively explored avenue in the search for treatments for diseases associated with aberrant protein aggregation and degradation. A striking number of proteins involved in a wide spectrum of diseases undergo sumoylation. In this section, we summarize known impacts of sumoylation in PQC-associated human diseases, with an emphasis on the distinct effects of SUMO1 and SUMO2/3 on the relevant pathogenic proteins.

### 13.1. Cystic Fibrosis

One of the earliest understandings of the roles of sumoylation in PQC-associated disease arose from its regulation of the cystic fibrosis transmembrane conductance regulator (CFTR) [86]. Regulation of CFTR is also one of the first demonstrations that modification by different SUMO paralogues can lead to divergent fates of substrate proteins (Figure 6). CFTR is a cAMP-regulated anion channel protein located at the apical membrane of secretory epithelial cells [121]. Mutations in CFTR underlie cystic fibrosis, in which the most common mutation is deletion of a phenylalanine residue at position 508 (F508del), leading to misfolding and enhanced proteasomal degradation of the protein [121]. CFTR is modified by both SUMO1 and SUMO2/3, but conjugation by each paralogue leads to opposing outcomes. Specifically, it was shown that Hsp27 cooperates with Ubc9 to selectively conjugate mutant CFTR with poly-SUMO2/3 chains, which are subsequently recognized by RNF4 and degraded by the proteasome [122]. On the other hand, SUMO E3 ligase PIAS4 modifies CFTR with SUMO1, which increases abundance and stability of both wild-type and F508del mutant CFTR proteins [86]. It is therefore possible that SUMO1 modification promotes CFTR stability during biogenesis, whereas selective poly-SUMO2/3 modifications on mutant CFTR proteins induce RNF4-mediated degradation. The stabilizing effect of SUMO1 modification on mutant CFTR may also function to compensate for insufficient CFTR abundance in airway cells due to rapid degradation of mutant proteins, providing a partial rescue of anion channel function.

### 13.2. Cardiovascular Disease

Heart disease is the leading cause of death worldwide [123]. Cardiomyocytes have poor regenerative ability and are therefore more sensitive to proteomic stress than other somatic cells. Accumulating evidence indicates that disturbance of proteostasis in cardiomyocytes is a contributing factor to the development of cardiovascular disease [124,125,126]. Sumoylation carries out critical functions in both developing and adult hearts. For example, SUMO1 knockout mice exhibit increased congenital heart defects [28], while SUMO1 overexpression is protective against heart failure in both murine and swine animal models [127,128]. In contrast, enhanced SUMO2 conjugation increases apoptosis and aggravates cardiomyopathy and heart failure [129]. Recent studies had also suggested a role for sumoylation in maintaining cardiac proteostasis [130,131]. As evidence, overexpression of the SUMO E2 conjugating enzyme Ubc9 reduces pathogenic protein aggregation and improves cardiac function in a desmin-related cardiomyopathy model [130]. In this model, Ubc9 overexpression prevents aggregation and enhances autophagic clearance of a mutant small heat-shock protein chaperone, alpha-crystallin B chain (CryAB), which otherwise compromises autophagic and proteasomal degradation in cardiomyocytes. In a successive study, the authors further showed that overexpressing Ubc9 enhances overall autophagy and imparts protective functions in rodent cardiomyocytes [132]. Recent proteomic analysis also revealed dynamic changes in sumoylation levels of UPS components in mouse hearts during ischemia reperfusion, suggesting a role for sumoylation in cardiac proteostasis following ischemic injury [133]. Currently, roles of sumoylation in cardiac proteostasis and heart diseases were mainly derived from gain or loss-of-function studies using animal models. The underlying molecular mechanisms by which sumoylation impacts cardiac PQC, as well as effects of different SUMO paralogues in heart physiology and pathology, remain poorly understood.

### 13.3. Neurodegeneration

Maintaining proteostasis is particularly challenging in neurons due to their unique extended morphology. Furthermore, neuronal cells are terminally differentiated and cannot dilute proteotoxic stress by division, leading to higher risks of accumulating damaged proteins [90]. In fact, many neurodegenerative diseases are considered protein misfolding disorders. Modulating PQC is therefore a potential therapeutic strategy to limit the progression of these currently incurable diseases. Sumoylation is implicated in nearly all forms of neurological disorders, including Parkinson’s disease (PD) [134], amyotrophic lateral sclerosis (ALS) [135], Huntington’s disease (HD) [136], spinocerebellar ataxia (SCA) [35], Alzheimer’s disease (AD) and dentatorubral-pallidoluysian atrophy (DRPLA) [137]. However, the overall impacts of sumoylation vary in each disease, and individual studies sometimes yield conflicting observations. A comprehensive list of known impacts of sumoylation in neurodegenerative diseases is provided in Table 3.

The divergent outcomes of sumoylation in neurodegenerative diseases may be attributed to different cellular localizations of pathogenic proteins, sumoylation at different conjugation sites, competitive or cooperative regulation with other PTMs, modification by different SUMO paralogues and recruitment of distinct downstream effectors. Overall, as discussed in the previous sections, we note that SUMO2/3 modification on pathogenic proteins generally promotes their degradation through recruitment of STUbLs and induction of ubiquitin-mediated degradation, whereas SUMO1 modification promotes substrate solubility and/or stability in a context-dependent manner. As evidence, SUMO1 modification promotes solubility of α-synuclein (PD) and huntingtin (HD) [7,136], while SUMO2 modification of ataxin-1 and ataxin-7 (SCA type 1 and 7, respectively) promotes degradation by recruiting RNF4 [35,138]. On the other hand, SUMO1 modification stabilizes Tau (AD) and SOD1 (ALS), which seems to have an overall deleterious effect and increases cytotoxicity of these proteins [139,140]. Effects of SUMO2/3 conjugation to these proteins are less clear. Regarding impacts on protein aggregation, SUMO1 modification of α-synuclein and amyloid-beta ameliorates aggregation while modification of SOD1 exacerbates aggregation [7,140,141]. Of note, altered solubility and aggregation status of pathogenic proteins may result in different pathological outcomes under different contexts. For some proteins, such as α-synuclein and androgen receptor, enhanced solubility prevents formation of cytotoxic aggregates, therefore leading to protective effects [73,142]. While in another scenario, sequestration of aberrant, cytotoxic proteins into cellular inclusions may be a protective mechanism in cells to limit toxicity. In this case, increased solubility of pathogenic proteins may lead to a more deleterious effect, as shown by some studies of huntingtin in Huntington’s disease [143,144].

Apart from affecting specific pathogenic proteins, recent studies also suggested a role for the nuclear SUMO2/3–RNF4 pathway in regulating cytosolic stress granule (SG) dynamics and recruitment of disease-associated proteins into these subcellular structures [145,146]. SGs are membraneless organelles formed in response to various stress insults. Accumulation of persistent SGs and a liquid-to-solid transition of these dynamic organelles, potentially from aggregation of SG components, are observed in ALS and other neurodegenerative diseases [147,148]. It has been shown that an intact SUMO2/3-RNF4 pathway is required for proper SG resolution and limits the compartmentalization of an ALS-associated FUS mutant into SGs [146]. RNA-binding proteins are extensively sumoylated and are also sequestered into SGs under stress [146,149]. The SUMO2/3–RNF4 mediated nuclear PQC pathway is therefore proposed to be a surveillance mechanism for aggregation-prone RNA binding proteins, including TDP-43 and FUS. Defects in this pathway may contribute to proteotoxic stress and interfere with proper SG dynamics and functions [146].

In general, sumoylation seems to impart an overall detrimental effect under contexts of Huntington’s disease, Alzheimer’s disease, DRPLA, and ALS. While in Parkinson’s disease and various types of SCA, sumoylation seems to exhibit a generally protective effect. Current understandings of the effects of SUMO1 and SUMO2/3 on specific pathogenic proteins and in particular diseases have been gained from individual studies using different organisms. To potentially target sumoylation for the development of targeted therapy against these diseases, it would be necessary to systematically analyze the biomolecular consequences of SUMO1 versus SUMO2/3 conjugation on relevant pathogenic proteins and in specific cellular pathways.

**Table 3 cells-13-00008-t003:** Known impacts of sumoylation in PQC-associated disease.

Disease	Affected Protein	Involved Sumoylation Component	Modified Sites	Conclusions	Overall Outcome of Sumoylation	Experimental Models	Reference
CF	CFTR	SUMO2/3 Ubc9 RNF4	-	Hsp27 promotes SUMO2/3 modification of CFTR^F508del^ and enhances its degradation through the RNF4-mediated pathway	-	HEK293 cells; Calu-3 cells	Ahner et al., 2013 [122]
	CFTR	PIAS4 SUMO1	K377, K447, K1199, K1468	PIAS4 increases WT and F508del mutant CFRT biogenesis and stability. For CFRT^F508del^, PIAS4 promotes SUMO1 conjugation and reduces SUMO2/3 conjugation	Potentially protective	CFBE airway cells	Gong et al., 2018 [86]
Cardiomyopathy	-	Ubc9	-	Ubc9 overexpression enhances UPS function in cardiomyoctes and decreases protein aggregation	Protective	Mouse models; NRVCs	Gupta et al., 2014 [130]
	-	Ubc9	-	Ubc9 overexpression enhances cardiac autophagy and improves cardiac function	Protective	Mouse models; NRVCs	Gupta et al., 2016 [132]
HD	Huntingtin (Htt)	SUMO1	K6, K9, K15	SUMO1 modification increases Htt stability while reducing aggregation	Toxic	Striatal nerve cells; fly HD model	Steffan et al., 2004 [143]
	Htt	SUMO1, SUMO2 PIAS1	K6, K9	PIAS1 promotes sumoylation of Htt, which decreases protein solubility; reduction of PIAS1 is neuroprotective in a fly HD model	Toxic	HeLa cells; human brain tissues; fly HD model	O’Rourke et al., 2013 [150]
	Htt	PIAS1	-	Reducing PIAS1 levels ameliorates Htt-associated phenotypes in R6/2 mice	Toxic	Mouse models	Ochaba et al., 2016 [151]
	Htt	SUMO1 Rhes	-	SUMO1 knockout increases autophagic activity, reduces soluble Htt levels, alters distribution of Htt aggregates and attenuates disease phenotypes in HD mice	Toxic	Mouse- and patient-derived cell lines; mouse models	Ramırez-Jarquın et al., 2022 [136]
AD	Amyloid-β (Aβ)	SUMO3	K587, K595	Poly-SUMO3 modification reduces Aβ production while mono- and hyposumoylation increase Aβ generation	-	293T cells; SK-N-MC neuroblastoma cells	Li et al., 2003 [152]
	Aβ	SUMO3	-	SUMO3 affects APP processing and increases Aβ production independent of conjugation	-	HEK293 cells	Dorval et al., 2007 [153]
	Aβ	SUMO1 SUMO2 Ubc9	K587 K595	Sumoylation decreases Aβ aggregation	Protective	HeLa cells	Zhang and Sarge, 2008 [141]
	Aβ	SUMO1	-	SUMO1-APP transgenic mice shows normal APP processing but increased insoluble Aβ and plaque density at later ages, which may be due to impaired Aβ clearance	Toxic	Transgenic mice	Knock et al., 2018 [154]
	BACE1	Overall sumoylation	K275, K501	Sumoylation at K501 enhances BACE1 stability and enzymatic activity, leading to increased Aβ toxicity	Toxic	HEK293T cells; mouse neuroblastoma cells	Bao et al., 2018 [155]
	Tau	SUMO1	K340	SUMOylation of Tau reciprocally stimulates its phosphorylation, reduces ubiquitylation, and inhibits degradation	Toxic	HEK293 cells; rat hippocampal nuerons	Luo et al., 2014 [139]
PD	α-synuclein Tau	SUMO1	α-synuclein: K96, K102; Tau: K340	Tau and α-synuclein are both preferentially modified by SUMO1. Hyperphosporylation and proteasome inhibition enhance Tau sumoylation	-	HEK293 cells	Dorval and Fraser, 2006 [134]
	α-synuclein	SUMO1	-	Proteasome inhibition enhances α-synuclein sumoylation; SUMO1 is found in Lewy bodies in PD and DLB patient brains	-	COS-7 cells; patient samples	Kim et al., 2011 [156]
	α-synuclein	SUMO1, SUMO2	K96, K102	Sumoylation promotes α-synuclein solubility and inhibits aggregation	Protective	In vitro assays; HEK293 cells; rat PD model	Krumova et al., 2011 [7]
	α-synuclein	Polycomb protein2 (Pc2) SUMO1	-	Pc2 promotes α-synuclein sumoylation and aggregation, which reduces cell death in response to staurosporine	Protective	COS-7 cells; HEK293 cells	Oh et al., 2011 [157]
	α-synuclein	SUMO1, SUMO3	K102 K96	Inhibitory effect of sumoylation on α-synuclein aggregation is site- and SUMO isoform-dependent. SUMO1 is a better inhibitor than SUMO3	Protective	In vitro assays	Abeywardana and Pratt, 2015 [73]
	α-synuclein	SUMO2	-	Sumoylation regulates sorting of α-synuclein into extracellular vesicles in an ESCRT-dependent manner	-	Oli-neu and N2a cells; HEK293 cells; mouse neurons	Kunadt et al., 2015 [158]
	α-synuclein	PIAS2 Overall sumoylation	-	PIAS2 promotes sumoylation of α-synuclein, which counteracts ubiquitin-mediated degradation and promotes aggregation. PD brains show increased levels of PIAS2 and sumoylated α-synuclein	Toxic	HEK293 cells; In vitro assays; PD patient samples	Rott et al., 2017 [159]
	α-synuclein Tau	TRIM28	-	TRIM28 depletion in adult mice reduces levels of α-synuclein and Tau without exhibiting behavioral or pathological phenotypes	Toxic	HEK293 cells; mouse models	Rousseaux et al., 2018 [160]
	DJ-1	SUMO1, PIASx*α*, PIASy	K130	Sumoylation is essential for DJ-1 functions. DJ-1 mutants that are improperly sumoylated are less soluble and partially localized to mitochondria	Protective	Human H1299, ME180, 293T and HeLa cells; mouse NIH3T3 cells	Shinbo et al., 2006 [161]
ALS	SOD1	SUMO1	K75	Sumoylation of both WT and mutant SOD1 at K75 increases its stability and promotes aggregation	Toxic	HEK293 cells	Fei et al., 2006 [140]
	SOD1	SUMO3	K75	Modification of mutant SOD1 proteins by SUMO3, but not SUMO1, increases its stability and aggregation	Toxic	NSC34 cells; CHO cells; HEK293 cells	Niikura et al., 2014 [162]
	SOD1	SUMO1, overall sumoylation	K75	Inhibiting sumoylation on K75 prevents aggregation of SOD1 pathogenic mutants. SUMO1 colocalizes with SOD1 aggregates	Toxic	NSC34 cells	Dangoumau et al., 2016 [163]
	SOD1	SUMO3, PIAS ligases, SENP1,	K75	PIAS increases SOD1 aggregation, while SENP1 decreases SOD1 aggregation	Toxic	NSC34 cells	Wada et al., 2020 [135]
	TDP-43 TDP-S6	SUMO2/3	K136	TDP-43 overexpression increases SUMO2/3 levels in the insoluble proteome; SUMO2/3 and poly-ubiquitin are found in TDP-43 aggregates	-	Mouse primary neurons; HEK293 cells	Seyfried et al., 2010 [164]
	TDP-43	SUMO1, overall sumoylation	K136	Inhibiting sumoylation by anacardic acid reduces TDP-43 aggregation; mutating K136 reduces TDP-43 cytosolic localization	Toxic	NSC34 cells; HEK293T cells; mouse primary motor neurons.	Maurel et al., 2020 [165]
	TDP-43	Overall sumoylation	K136	Sumoylation affects TDP-43 exon skipping activity and nucleo-cytoplasmic trafficking; mutating K136 reduces TDP-43 recruitment into stress granules	-	SK-N-BE cells; HEK293T cells	Maraschi et al., 2021 [166]
	Androgen receptor (AR)	SUMO3	K385, K518	Sumoylation of PolyQ-expanded AR reduces its aggregation, increases solubility and reduces toxicity	Protective	HeLa cells	Mukherjee et al., 2009 [142]
	-	Sae1, Ubc9 E3 ligases	-	Sumoylation regulates stress granule disassembly, and the relevant mechanism is impaired in C9orf72-associated ALS	Protective	U2OS cells; fly ALS model	Marmor-Kollet et al., 2020 [145]
	TDP-43, FUS, RBPs	RNF4	-	The nuclear SUMO2/3–RNF4 pathways is required for proper granule resolution; RNF4 limits recruitment of a ALS-associated FUS mutant into SGs	-	HeLa cells; U2OS cells	Keiten-Schmitz et al., 2020 [146]
SCA Type 1	Ataxin-1, Htt, TDP-43	PML, SUMO2/3, RNF4	-	PML mediates SUMO2/3 conjugation to nuclear pathogenic proteins, which promotes ubiquitination and proteasomal degradation by recruiting RNF4	Protective	In vitro assays; HeLa cells; SCA1 mouse model	Guo et al., 2014 [35]
	Ataxin-1	SUMO1 Ubc9	Multiple sites	SUMO1 or Ubc9 over-expression increases ataxin-1 aggregation; oxidative stress and the JNK pathway impact ataxin-1 sumoylation	Toxic	BOSC23 cells;	Ryu et al., 2010 [167]
SCA Type 3	Ataxin-3	SUMO1 SUMO2	K356	Sumoylation of ataxin-3 at K356 decreases amyloid fibril formation and increases its affinity to VCP/p97	Protective	In vitro assays; COS-7 cells	Almeida et al., 2015 [168]
SCA Type 7	Ataxin-7	SUMO2 RNF4	-	SUMO2 modification promotes ataxin-7 degradation by recruiting RNF4	Protective	HEK293, HeLa, MCF7 cells; mouse model; SCA7 patient samples	Marinello et al., 2019 [138]
	Ataxin-7	SUMO1 SUMO2	K257	Sumoylation increases solubility of polyQ extended ataxin-7, decreasing aggregation propensity and cellular toxicity	Protective	COS-7 cells; patient samples; transgenic mice	Janer et al., 2010 [169]
DRPLA	Atrophin-1	SUMO1	K111	SUMO1 colocalizes with inclusions in DRPLA brains and increases intranuclear aggregation and cell death	Toxic	DRPLA brain tissues; PC12 cells	Terashima et al., 2002 [170]

CF: cystic fibrosis; HD: Huntington’s disease; AD: Alzheimer’s disease; PD: Parkinson’s disease; ALS: amyotrophic lateral sclerosis; SCA: spinocerebellar ataxia; DRPLA: dentatorubral-pallidoluysian atrophy.

## 14. Summary and Future Direction

As a highly dynamic and versatile PTM, sumoylation can affect cellular PQC in complex ways and on multiple levels. First, sumoylation can regulate expression of PQC pathway components at the level of gene transcription. This function is supported by proteomic and gene ontology analyses showing that transcription factors are among the most frequently detected sumoylation targets [171]. Previous RNA sequencing analysis in our lab using SUMO1 and SUMO2/3 knockout cell lines further revealed unique effects of each paralogue on the transcriptome, including altered expression of multiple chaperones and heat shock factors [17]. Second, SUMO can act on target proteins directly to regulate their solubility and stability, achieved through both direct conjugation and noncovalent SIM-association. This function is exemplified by protein degradation through the SUMO2/3–STUbL–UPS pathway and SUMO1-mediated changes in substrate solubility and aggregation propensities [6,35]. Furthermore, SUMO can also impact cellular PQC by directly regulating factors within PQC pathways, such as molecular chaperones and the proteasome [10,172]. Apart from cooperatively mediating protein degradation, sumoylation and ubiquitylation can also compete with each other for conjugation at the same lysine sites, resulting in altered fates of target proteins. Given the relatively low steady state levels of sumoylation, effective competition would depend upon a select “active” pool of substrate that is prone to both modifications. This complexity offers the opportunity for unique molecular interventions that could be used to treat the numerous human diseases with links to sumoylation.

A well-characterized therapeutic strategy involving sumoylation is the treatment of acute promyelocytic leukemia (APL). Arsenic trioxide has emerged as an effective treatment for APL by inducing degradation of the PML-RARα oncogenic fusion protein [173]. Arsenic trioxide specifically induces sumoylation of PML-RARa and leads to its degradation through the RNF4–UPS pathway [113,174]. Previously, ginkgolic acid, a natural compound extracted from Ginkgo biloba leaves that inhibits sumoylation, was found to promote clearance of α-synuclein aggregation and could therefore have therapeutic value in treating Parkinson’s disease [175]. Other selective inhibitors that target the sumoylation pathway have also been developed, including the SUMO E1 inhibitor, TAK-981, which is currently in clinical trials to treat cancer [176]. TAK-981 and other sumoylation inhibitors developed to date target either the E1 or E2 enzyme and therefore reduce functions of all SUMO paralogues. Future development of strategies to target individual SUMO paralogues or paralogue-specific functions would provide a new avenue for the development of more precise therapies with less adverse effects. Targeted protein degradation is an emerging and robustly growing field that harnesses the ubiquitin–proteasome system to tackle pathogenic proteins that have been previously undruggable [177]. Sumoylation pathway components, such as specific E3 ligases and SENPs, may also be employed and engineered to manipulate the function and degradation of disease-causing protein targets.

## Figures and Tables

**Figure 1 cells-13-00008-f001:**
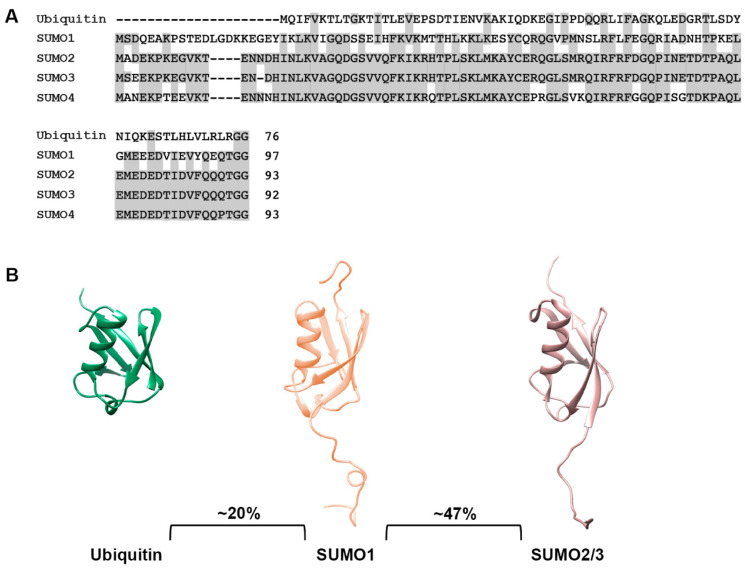
Sequence and structural comparison of ubiquitin and SUMO paralogues. (**A**) Protein sequence comparison of ubiquitin and SUMO paralogues. Analysis was performed using the COBALT multiple alignment tool. (**B**) Comparison of ubiquitin, SUMO1 and SUMO2/3 structures. Protein structures were generated using PDB files and the PyMOL program.

**Figure 2 cells-13-00008-f002:**
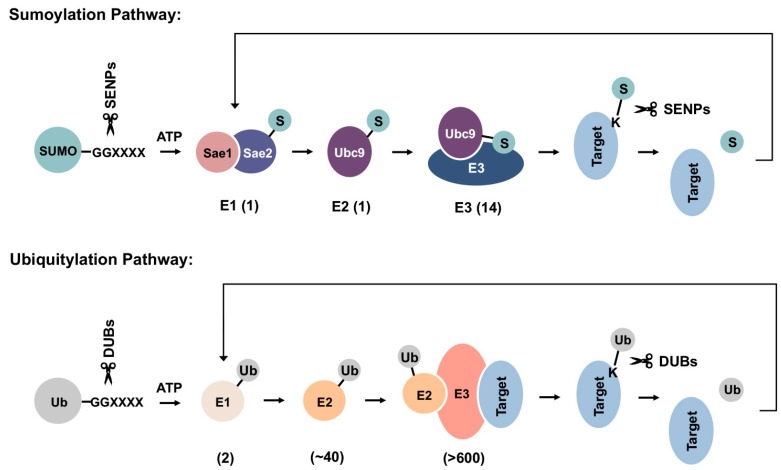
Schematics of sumoylation and ubiquitylation pathways. SUMO precursors are first cleaved by sentrin-specific proteases (SENPs) to expose a C-terminal di-glycine motif required for ATP-dependent activation by the E1 enzyme heterodimer. Activated SUMOs can then be transferred to the E2 conjugating enzyme, Ubc9, through a thioester bond. Finally, SUMOs are conjugated to a lysine residue within target protein, either directly or with the help of an E3 ligase. Sumoylation can be reversed by SUMO-specific isopeptidases, which deconjugate and recycle SUMOs from substrate proteins. Ubiquitylation happens in a similar manner but employs distinct sets of enzymes in each step. DUBs: deubiquitinases. Numbers shown below each reaction are the number of related enzymes identified in humans.

**Figure 3 cells-13-00008-f003:**
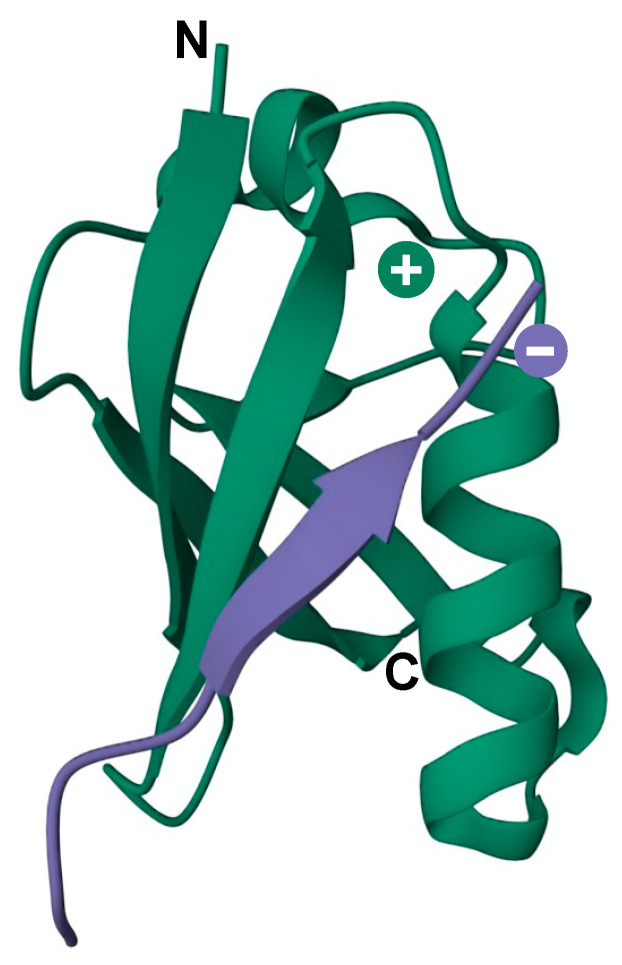
3D model of the secondary structure of a SIM motif (purple) in contact with the SUMO surface (green). Negative and positive charges indicate the SIM acidic region and SUMO basic patch, respectively. N and C indicate the N- and C-terminus of SUMO protein. PDB: 6V7P.

**Figure 4 cells-13-00008-f004:**
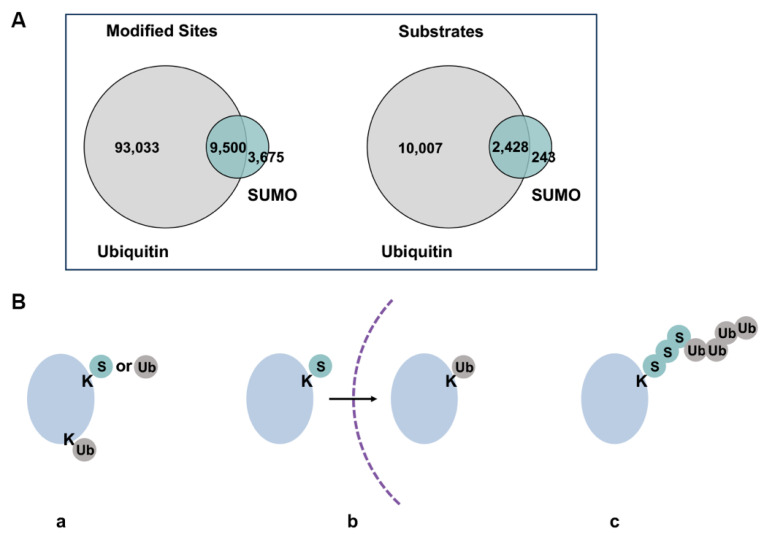
Overlaps of sumoylated and ubiquitylated substrates. (**A**) Overlap of currently identified ubiquitin and sumoylation sites by proteomic analyses. Figure was adapted from Trulsson and Vertegaal, 2021 [82]. Difference in numbers of identified sites and substrates may be due to varied depths of analyses for the two modifications. (**B**) Different co-modification patterns of SUMO and ubiquitin, where a protein can be modified by SUMO or ubiquitin on the same or different lysine residues (**a**), sequentially modified by SUMO or ubiquitin during biological processes or at different subcellular localization (**b**), modified by SUMO-ubiquitin hybrid chains (**c**). Dash line represents intracellular membranes.

**Figure 5 cells-13-00008-f005:**
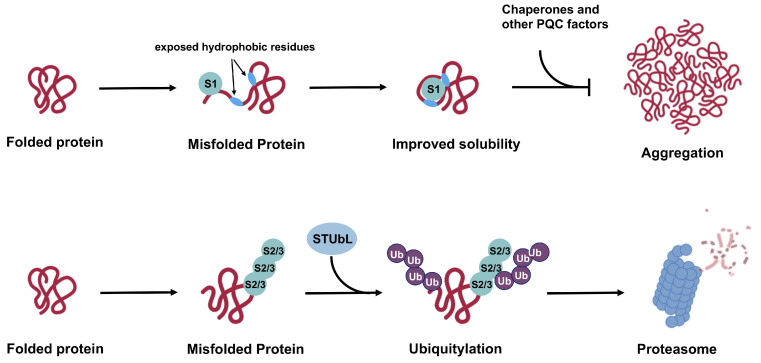
Proposed mechanisms for SUMO1 versus SUMO2/3 functions in cellular PQC. While poly-SUMO2/3 modification works in concert with STUbL and the UPS to promote substrate degradation, SUMO1 conjugation promotes solubility of misfolded substrates and prevents aggregation in cooperation with cellular chaperones and other PQC factors. Effects of SUMO1 on protein solubility and aggregation status may lead to protective or deleterious cellular outcome based on different contexts and specific substrates. In other cases, SUMO1 modification may counteract functions of poly-SUMO2/3, leading to stabilization of substrate proteins.

**Figure 6 cells-13-00008-f006:**
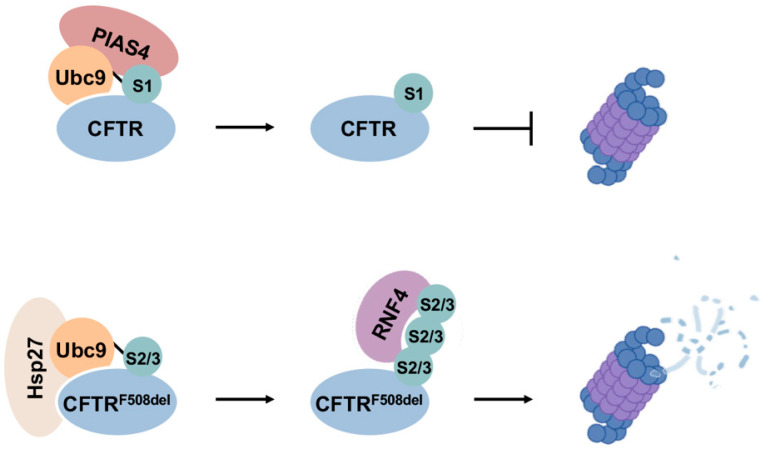
Divergent effects of SUMO1 and SUMO2/3 in CFTR biogenesis and degradation. SUMO1 modification of CFTR promotes protein stability during biogenesis, while selective modification of mutant CFTR by poly-SUMO2/3 enhances degradation through the RNF4–UPS degradation pathway.

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
