# Peer review of "Paralogue-Specific Roles of SUMO1 and SUMO2/3 in Protein Quality Control and Associated Diseases"

_cells, 2023, doi:10.3390/cells13010008_

Round 1
Reviewer 1 Report
Comments and Suggestions for Authors
The manuscript from Wang and Matunis reviews the SUMO protein family, the sumoylation pathway and SUMO modifications. The authors specifically focus on SUMO1 and SUMO2/3 discussing their role in maintaining proteostasis and the consequent implication in human diseases.
The manuscript is well structured and may be useful as an update on SUMOs and their functions in protein quality control. I just have few minor suggestions for its improvement.
line 120. It is reported that about 10 SUMO E3 ligases have been characterized. The precise number should be indicated and maybe the ligases should be specified either in the text or in a Table.
Figure 2. Please, specify SENPs in the legend. As before, the precise number of ligases should be indicated.
Figure 3. The 3D model of SIM contacts with SUMO is interesting. I would include this model in Figure 1, close to ubiquitin and SUMO models.
Table 1. The table is informative but difficult to read. Please, modulate column width, font size and/or paragraph size to avoid overlaps.
Table 2. The Table summarize information on sumoylation impact in neurodegenerative diseases. Please, specify disease abbreviations in a footnote. DRPLA should be also described in the text.
Finally, I would include in the Table information on cystic fibrosis and cardiovascular disease as reported in the text.
Author Response
Dear Reviewer,
Thank you for your time and effort that has been dedicated to providing valuable feedback to improve our manuscript. We are submitting a revised manuscript with tracked changes for your review. A point-by-point response to your comments and suggestions and a summary of the modifications made to the manuscript are as follows:
(1) In line 120, it is reported that about 10 SUMO E3 ligases have been characterized. The precise number should be indicated and maybe the ligases should be specified either in the text or in a Table.
We have added a table (page 5) listing human SUMO E3 ligases characterized to date and updated Figure 2 accordingly.
(2) Figure 2. Please, specify SENPs in the legend. As before, the precise number of ligases should be indicated.
We have specified SENPs in the legend and related text (line 164-167 in revised manuscript) and indicated the number of ligases.
(3) Figure 3. The 3D model of SIM contacts with SUMO is interesting. I would include this model in Figure 1, close to ubiquitin and SUMO models.
Thank you for this suggestion. We think it is best to not combine Figure 3 with Figure 1, as this would affect coordination of the text with appearance of the figures.
(4) Table 1. The table is informative but difficult to read. Please, modulate column width, font size and/or paragraph size to avoid overlaps.
We have modified the table (current Table 2) to make it easier to read.
(2) Table 2. The table summarize information on sumoylation impact in neurodegenerative diseases. Please, specify disease abbreviations in a footnote. DRPLA should be also described in the text. Finally, I would include in the Table information on cystic fibrosis and cardiovascular disease as reported in the text.
We have included disease abbreviations in the footnote of current Table 3 and added descriptions on DRPLA in the corresponding text. As suggested, we have also included information on cystic fibrosis and cardiovascular diseases in the table.
We once again thank you for providing us with valuable suggestions to improve the quality of our review. We hope that the revisions outlined above address your main concerns and that the manuscript is now acceptable for publication in Cells.
Sincerely,
Michael J. Matunis
Reviewer 2 Report
Comments and Suggestions for Authors
This manuscript comprehensively examines the distinctive roles played by each SUMO paralog, placing particular emphasis on their involvement in protein quality control pathways and the dysregulations observed in pathological conditions. It begins by offering a thorough exploration of the various SUMO proteins and describes the enzymology governing the SUMOylation pathway. Then, it delineates the diverse SUMO target proteins, highlighting the proteomic methodologies employed for their identification. The focus then shifts to elucidating the impact of SUMO isoforms on protein quality control (PQC) within both the nucleus and cytosol, together with the molecular interconnections with the ubiquitin pathway. The review ends with an insightful section addressing the influence of SUMO-1 and SUMO-2/3 on proteins associated with PQC-related pathologies. This review is very well written and provides an in-depth perspective on the still poorly characterized specificities of the SUMO paralogs.
Minor comments:
- In the text (line 111), it is indicated AOS1/UBA2 and on the figure 2, it is indicated SAE1/uba2. The official nomenclature should be used (Sae1/Uba2).
- Line 182: I would rather refer to « proteotoxic stress » than « proteomic stress » when refering to proteasome inhibition
- Line 256 suggests that no comparison was made between SUMO-1 and SUMO-2/3 conjugated proteome. However, the study cited at the beginning of the paragraph (ref 68) indeed gives an idea of the common and different targets between SUMO-1 and 2 targets. Other studies using SUMO1 or SUMO-2/3 immunoprecipitations have also provided insight on the paralog specificy (e.g PMID 33480129, 30926672, 37462077).
- It could be mentioned that the overexpression of SUMO, which was used in various proteomic approaches, makes that the paralog specificity is probably lost.
- Line 431: From what I understand, it is suggested there that SUMO-1 modification would favor the recruitment of p97. It has however been suggested recently that p97 complex might rather be recruited to targets (at least PML) via StuBL-mediated ubiquitylation rather than direct binding to SUMOylated PML (PMID 36880596). Considering that PML was found mostly conjugated by SUMO-1 rather than SUMO-2/3 upon arsenic treatment, StubL could indeed bind multi-mono sumoylated targets with SUMO1 and induce their ubiquitylation. I’m therefore not completely convinced that SUMO1 conjugation necessarily blocks ubiquitylation (line 441). This possibility could be discussed.
Author Response
Dear Reviewer,
Thank you for your time and effort that has been dedicated to providing valuable feedback to improve our manuscript. We are submitting a revised manuscript with tracked changes for your review. A point-by-point response to your comments and suggestions and a summary of the modifications made to the manuscript are as follows:
(1) In the text (line 111), it is indicated AOS1/UBA2 and on the figure 2, it is indicated SAE1/uba2. The official nomenclature should be used (Sae1/Uba2).
We have standardized all references to the SUMO E1 as Sae1/Sae2 across the manuscript.
(2) Line 182: I would rather refer to « proteotoxic stress » than « proteomic stress » when referring to proteasome inhibition.
We have corrected the relevant texts to “proteotoxic stress”.
(3) Line 256 suggests that no comparison was made between SUMO-1 and SUMO-2/3 conjugated proteome. However, the study cited at the beginning of the paragraph (ref 68) indeed gives an idea of the common and different targets between SUMO-1 and 2 targets. Other studies using SUMO1 or SUMO-2/3 immunoprecipitations have also provided insight on the paralog specificity (e.g PMID 33480129, 30926672, 37462077)
Our intent was to emphasize that in comparison to the significant efforts to identify SUMO2-modified proteins, proteomic analysis of SUMO1-modifed proteins is more limited. Moreover, systematic comparison of SUMO1- and SUMO2-modified proteins using more recently developed and sensitive proteomic strategies in the same study is also limited. We have modified the text to better reflect these views (line 264-266 in revised manuscript).
(4) It could be mentioned that the overexpression of SUMO, which was used in various proteomic approaches, makes that the paralog specificity is probably lost.
We appreciate the suggestion and have included this point in the text (line 266-268).
(5) Line 431: From what I understand, it is suggested there that SUMO-1 modification would favor the recruitment of p97. It has however been suggested recently that p97 complex might rather be recruited to targets (at least PML) via StuBL-mediated ubiquitylation rather than direct binding to SUMOylated PML (PMID 36880596). Considering that PML was found mostly conjugated by SUMO-1 rather than SUMO-2/3 upon arsenic treatment, StubL could indeed bind multi-mono sumoylated targets with SUMO1 and induce their ubiquitylation. I’m therefore not completely convinced that SUMO1 conjugation necessarily blocks ubiquitylation (line 441). This possibility could be discussed.
We did not intend to suggest that p97 interacts exclusively with SUMO1 and recognize the role of RNF4 and poly-SUMO2/3 chains in recruiting p97. To alleviate the confusion, we have removed the reference of p97 (line 454) and included discussion of possible function for multi-mono SUMO1 modification on recruiting STUbLs in the text (line 466-471 in revised manuscript).
We once again thank you for providing us with valuable suggestions to improve the quality of our review. We hope that the revisions outlined above address your main concerns and that the manuscript is now acceptable for publication in Cells.
Sincerely,
Michael J. Matunis
Reviewer 3 Report
Comments and Suggestions for Authors
This is an excellent review about the protein post-translational modification SUMO, comparing the different SUMO family members SUMO1 and SUMO2/3 and their different roles in protein quality control. It is a very thorough piece of work and is the first review to focus on this specific topic in the field. I have a few minor suggestions as detailed below.
1. SUMO5 is possibly a pseudogene as mentioned. There is no evidence at the endogenous protein level that SUMO5 truly exits, so please mention that SUMO5 results were generated with exogenous SUMO5 and that evidence is lacking for the existence of endogenous SUMO5.
2. Currently the authors make fairly black and white distinction between SUMO2/3 coupled to STUbLs and subsequent degradation versus SUMO1 and no degradation. This is a bit too binary given the report from the Dohmen lab on the preference of the STUbL RNF111 for SUMO1 capped SUMO2/3 chains (Sriramachandran et al. 2019 Nature Comm). Although the authors do mention and cite this paper (lines 373-374), they don’t fully incorporate this finding throughout the manuscript. Please include this also in the summary parts of the manuscript and add this concept to figure 5 to provide a more balanced view of the different alternatives.
3. Please add the interesting concept described in lines 223-227 to Figure 3.
4. Line 492, please don’t limit this to the US only.
5. Some Greek symbols are currently included in bold, there is no need to do this in bold.
6. Line 572- exampled should be exemplified.
7. Lines 576 and 577 and elsewhere: Concerning competition between SUMO and ubiquitin for the same lysine residues, please add thoughts on stoichiometry; you would need a very high stoichiometry modification to ensure efficient competition.
8. Author contributions: please provide details using the author initials instead of X.X., Y.Y., and Z.Z.
9. Please consider including the following relevant papers:
Kumar et al. 2017 Nature Comm
Keiten-Schmitz et al. 2020 Mol Cell
Marmor-Kollet et al. 2020 Mol Cell
Author Response
Dear Reviewer,
Thank you for your time and effort that has been dedicated to providing valuable feedback to improve our manuscript. We are submitting a revised manuscript with tracked changes for your review. A point-by-point response to your comments and suggestions and a summary of the modifications made to the manuscript are as follows:
(1) SUMO5 is possibly a pseudogene as mentioned. There is no evidence at the endogenous protein level that SUMO5 truly exits, so please mention that SUMO5 results were generated with exogenous SUMO5 and that evidence is lacking for the existence of endogenous SUMO5.
We have modified the text to better reflect this point (line 83).
(2) Currently the authors make fairly black and white distinction between SUMO2/3 coupled to STUbLs and subsequent degradation versus SUMO1 and no degradation. This is a bit too binary given the report from the Dohmen lab on the preference of the STUbL RNF111 for SUMO1 capped SUMO2/3 chains (Sriramachandran et al. 2019 Nature Comm). Although the authors do mention and cite this paper (lines 373-374), they don’t fully incorporate this finding throughout the manuscript. Please include this also in the summary parts of the manuscript and add this concept to figure 5 to provide a more balanced view of the different alternatives.
We agree that there is not one simple black and white model to describe SUMO1 and SUMO2 functions and introduced the model to emphasize what we feel is one important defining difference. There will of course be exceptions and other functions. We have introduced additional discussions on possible exceptions to our proposed model in the text (line 466-471 in revised manuscript).
(3) Please add the interesting concept described in lines 223-227 to Figure 3.
We have included this concept in Figure 5.
(4) Line 492, please don’t limit this to the US only.
We have revised the text as suggested.
(5) Some Greek symbols are currently included in bold, there is no need to do this in bold.
We have unbolded the symbols.
(6) Line 572- exampled should be exemplified.
We have changed “exampled” to “exemplified”.
(7) Lines 576 and 577 and elsewhere: Concerning competition between SUMO and ubiquitin for the same lysine residues, please add thoughts on stoichiometry; you would need a very high stoichiometry modification to ensure efficient competition.
We have added a discussion of the issue of stoichiometry to the text (lines 645-647 in revised manuscript), suggesting that competition would require a select pool of “active” substrate available for both ubiquitin and SUMO modification.
(8) Author contributions: please provide details using the author initials instead of X.X., Y.Y., and Z.Z.
We have filled in information regarding author contributions.
(9) Please consider including the following relevant papers:
Kumar et al. 2017 Nature Comm
Keiten-Schmitz et al. 2020 Mol Cell
Marmor-Kollet et al. 2020 Mol Cell
We have incorporated the suggested papers into the text (line 415-416 and line 600-613) and Table 3.
We once again thank you for providing us with valuable suggestions to improve the quality of our review. We hope that the revisions outlined above address your main concerns and that the manuscript is now acceptable for publication in Cells.
Sincerely,
Michael J. Matunis